# Dolutegravir based antiretroviral therapy compared to other combined antiretroviral regimens for the treatment of HIV-infected naive patients: A systematic review and meta-analysis

**Mario Cruciani**[1]*, **Saverio G. Parisi**[2]

**1** Infectious Diseases Unit, AULSS 9 scaligera-Verona, Verona, Italy, **2** Department of Biotechnology, University of Padua, Padua, Italy

* crucianimario@virgilio.it

## Abstract

### Background

Numerous randomized clinical trials (RCTs) were conducted to evaluate dolutegravir based triple antiretroviral therapy (ART) compared to other triple antiretroviral regimens in naïve patients, and a summary of the available evidence is required to shed more light on safety and effectiveness issues.

### Methods

Systematic review and meta-analysis of RCTs comparing dolutegravir-containing ART to non-dolutegravir containing ART in HIV-infected naive patients. Primary outcomes: % of patients with viral load<50 copies/mL at 48 weeks, stratified according to baseline viral load levels (< or >100.000 copies/mL); overall rate of discontinuation and/or switching for any cause (virologic failure, clinical failure, adverse events). Measure of treatment effect: Risk Difference (RD) with 95% confidence intervals (CIs). The GRADE system was used to assess the certainty of the body of evidence,

### Results

We included 7 RCTs (13 reports, 6407patients) comparing dolutegravir containing to non-dolutegravir containing ART, both in combination with 2 NRTIs. Controls were raltegravir or bictegravir (3 RCTs), boosted atazanavir or darunavir (2 RCTs) or efavirenz (2 RCTs). Rates of patients with VL <50 copies/ml were higher in dolutegravir recipients compared to controls at 48 weeks (RD, 0.05; 95% CIs, 0.03/0.08, p = 0.0002) and 96 weeks (RD, 0.06; 95% CIs, 0.03/0.10, p<0.0001); the average benefit of using dolutegravir was particularly evident at 48 weeks in the subgroup of patients with high baseline viral load (RD, 0.10; 95% CIs, 0.05/0.15; p< 0.0001; GRADE assessment: "high certainty of evidence"). Overall rate of discontinuation were lower in dolutegravir compared to controls (RD,-0.03, 95% CIs

**Funding:** The authors received no specific funding for this work.

**Competing interests:** MC has received honoraria as a speaker and/or advisor from Abbott, Bayer, Cephalon, Gilead, Novartis and ViiV Healthcare. SGP has received research grants from Gilead Sciences, ViiV Healthcare, Abbott, MS&D, and honoraria as a speaker from MS&D, Gilead Sciences, ViiV Healthcare, Abbvie, Janssen. These activities were not related to their work with this review. This does not alter our adherence to PLOS ONE policies on sharing data and materials.

-0.05/-0.01; p = 0.007). No significant differences were observed in rates of discontinuation due to adverse events (RD, -0.02; 95% CIs, -0.05/0.00), virologic failure (RD, -0.01; 95% CIs, -0.02/0.01), and most common adverse events (GRADE assessment: from "very-low" to "moderate certainty of evidence")

## Conclusion

Starting treatment in naive patients with dolutegravir containing ART has an increased likelihood of achieving viral suppression in the comparison with non-dolutegravir containing ART. The average benefit is particularly evident in those with high baseline viral load.

## Introduction

The advent of antiretroviral therapy (ART) has made human immunodeficiency virus (HIV) infection a chronic manageable disease for many patients [1]. Individuals taking ART have now a wide range of therapeutic options and good prospects for long term survival. The current standard of care for HIV treatment is a three-drug regimen containing a nonnucleoside reverse transcriptase inhibitor (NNRTI), a protease inhibitor (PI), or an integrase strand transfer inhibitor (INSTI) plus two nucleoside/tide reverse transcriptase inhibitors (NRTIs), usually abacavir/lamivudine (ABC/3TC) or either tenofovir alafenamide/emtricitabine (TAF/FTC) or tenofovir disoproxil fumarate (TDF/FTC) [2]. Despite the undeniable success of antiretroviral therapy, limitations to safety and efficacy still persist. Moreover, in patients with baseline viral loads of greater than 100,000 copies/mL a slower rate of achieving viral suppression has been observed [2, 3].

Due to their favorable side effect profile, limited drug-drug interactions, and virologic potency, INSTI-based regimens are now among the recommended and preferred first-line ART for the treatment of HIV-1 infection in ART-naïve patients (a person with HIV who has never taken ART). Dolutegravir is a new-generation INSTI with distinct advantages compared with other available antiretroviral agents [4–7]. On the basis of data from in vitro studies and clinical trials in ART-naive patients, it is anticipated that, like dolutegravir, bictegravir has a high barrier to resistance. However, clinical data and experience with bictegravir are relatively limited at this time [2].

To assess the efficacy and safety of the combined ARTcontaining dolutegravir relative to other ART regimens not containing dolutegravir in naïve patients, we have conducted a systematic review and meta-analysis

## Material and methods

This review has been conducted according to the PRISMA statement for the reporting of systematic reviews and meta-analyses [8]. A protocol has been registered in PROSPERO, the prospective register of systematic reviews (Available from: http://www.crd.york.ac.uk/PROSPERO/display_record.php?ID=CRD42018104938).

### Search

The PubMed/MEDLINE, Embase, and Cochrane databases were systematically searched (up to April 2019) to identify randomized controlled trials (RCTs) evaluating efficacy and/or safety of DTG, EVG/c, RAL,BIC, ATV/r, DRV/r, EFV or RPV in treatment-naive HIV-1 patients.

PubMed and EMBASE search terms were "HIV-1 [mesh] OR HIV infections [mesh] AND ((dolutegravir OR GSK1349572) OR (efavirenz OR Sustiva OR Stocrin OR DMP-266) OR (raltegravir OR Isentress OR MK-0518) OR (elvitegravir OR GS-9137 OR JTK-303) OR (bictegravir OR GS-9883) OR (rilpivirine OR Edurant OR TMC 278) OR (darunavir OR Prezista OR TMC-114) OR (atazanavir OR Reyataz OR BMS-232632) OR (Atripla OR Quad OR Stribild OR Eviplera OR Complera))

We also searched reference lists of identified studies and major reviews, abstracts of conference proceedings, scientific meetings and clinical trials registries(www.clinicaltrials.gov).

## Types of study included

Randomized clinical trials (RCTs) conducted in adults HIV-infected naïve patients (> 18 years old) comparing antiretroviral regimen containing dolutegravir plus two NRTIs to regimens not containing dolutegravir (eg, NNRTI, PI, or other INSTI plus two NRTIs).

## Outcomes

### Primary outcomes.

- Virologic outcome: rates of patients with viral load (VL) below 50 copies/ml at 48 weeks and/or at 96 weeks

- Overall rate of discontinuation and/or switching for any cause (virologic failure, clinical failure, adverse events).

### Secondary outcomes.

- Rate of patients with any adverse event.

- Rates of adverse events requiring treatment interruption and/or switching.

- Occurrence of AIDS-defining illness

- Death (all cause)

- Change from baseline in CD4 count

## Data extraction

The two authors independently extracted data from the selected trials using standardized data extraction form. Agreement on the final assessment for each criterion was achieved by discussion. The following data were extracted: details of participants (number, setting, baseline characteristics by group), details of the study (study design; type and duration of follow up), details of ART regimen used, primary and secondary outcome descriptions and outcomes measures, number of withdrawals in each group with reasons. Data at 48 and 96 weeks were considered.

## Assessment of risk of bias and heterogeneity

We explored clinical heterogeneity (eg study setting, characteristics of participants) and assessed statistical heterogeneity using $Tau^2$, Cochran's Q and estimated this using the $I^2$ statistic, which examines the percentage of total variation across studies that is due to heterogeneity rather than to chance ($I^2 \leq 25$ suggest low heterogeneity; $I^2 > 50$, high heterogeneity) [9]. The risk of bias of each included study is assessed following the domain-based evaluation described in the Cochrane Handbook for Systematic Reviews of Interventions [10]. The Cochrane 'Risk

of bias' tool addresses six specific domains: sequence generation, allocation concealment, blinding, incomplete data, selective outcome reporting, and other issues relating to bias. We have presented our assessment of risk of bias using two 'Risk of bias' summary figures: 1) a summary of bias for each item across all studies; and 2) a cross-tabulation of each trial by all of the 'Risk of bias' item.

We used the principles of the GRADE (The Grading of Recommendations Assessment, Development and Evaluation) system to assess the quality of the body of evidence associated with specific outcomes, and constructed a 'Summary of findings' table using the software Review Manager, version 5.3 (available at https://community.cochrane.org/help/tools-and-software/revman-5/revman-5-download). The certainty of a body of evidence involves consideration of within-trial risk of bias (methodological quality), directness of evidence, heterogeneity, precision of effect estimates, and risk of publication bias. To inspect for publication bias visually we generated graphical funnel plots [10, 11].

### Strategy for data synthesis

We used aggregate data. The analysis was conducted on an "intention to treat" approach. Quantitative synthesis was used if the included studies were sufficiently homogenous.

When $I^2$ *values were* = 0, studies were pooled using a fixed-effect model. Where values of $I^2$ are greater than zero, both fixed and random effects analyses were performed and any differences in estimates of treatment effect considered. The fixed effect model was also considered as a sensitivity analysis for evaluating the possible bias effects of smaller studies. Potential sources of heterogeneity were explored by pre-specified subgroup analysis. We used risk difference as measure of effect. Review Manager 5.3 was used to analyze the data.

### Analysis of subgroups or subsets

We anticipated clinical heterogeneity in the effect of the intervention and we conducted, where possible, the following sub-group analyses:

- Virologic efficacy of dolutegravir and comparators analyzed according to baseline VL (eg, < 100,000 copies/ml or >100,000 copies/ml)

- Efficacy and safety data of dolutegravir analyzed according to the control intervention (eg, vs PI, vs NNRTI, or vs other INSTI).

  When sufficient trials were identified, we conducted a sensitivity analysis comparing the results using all trials as follow:

- those RCTs with high methodological quality (studies classified as having a 'low risk of bias' versus those identified as having a 'high risk of bias');

- Those RCTs that performed intention-to treat versus per-protocol analyses.

## Results

Electronic searches yielded 842 potentially relevant studies (Fig 1). 779 articles were excluded after preliminary screen and 63 were deemed potentially eligible and the full-text assessed.

Twenty-five studies were then excluded because they were not randomized. Of the 34 RCTs, 13 reports (corresponding to 7 studies) comparing dolutegravir-containing regimens to non-dolutegravir-containing regimens were conducted in naïve patients and were included in the qualitative and quantitative synthesis [12–24]. The main features of the included studies are summarized in Table 1. Controls were INSTI (raltegravir or bictegravir, 3 studies, 6

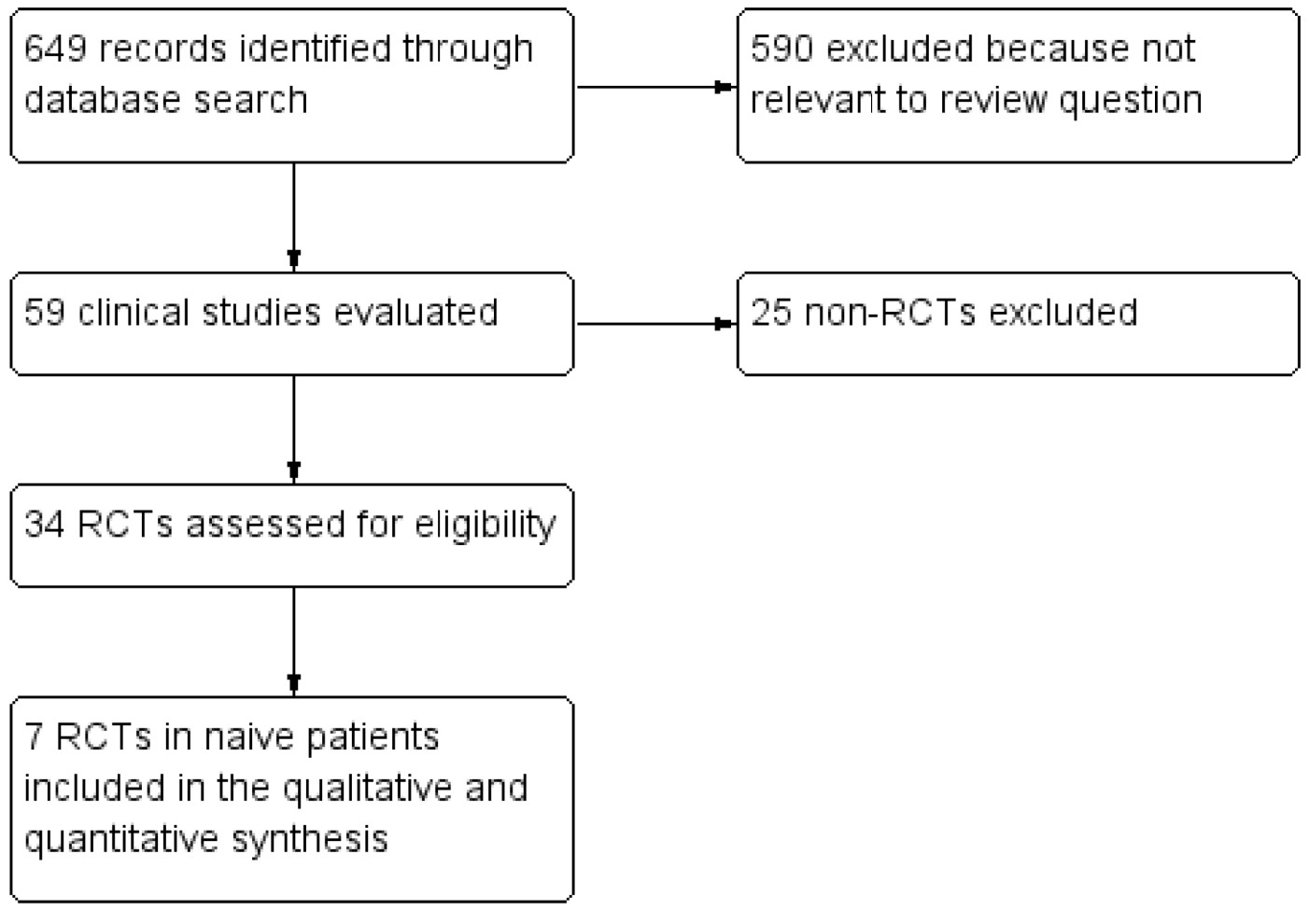

**Fig 1. Study flow diagram.**

reports) [14, 15, 18, 21, 23, 24], boosted PI (atazanavir or darunavir, 2 studies, 3 reports)[12, 13, 19] or NNRTI (efavirenz, 2 RCTs, 4 reports) [16, 17, 20, 22].

### Risk of bias in selected studies

Summary graphs of methodological quality items are presented in Figs 2 and 3. All the included studies were judged at low risk of selection, attrition and reporting bias. Three of the included studies were open-label, and we judged this domain as "high risk" of bias for these studies (ARIA, Flamingo, SPRING-1) [12, 13, 17, 19, 20]; however, masking probably has limited importance for more objective outcomes (as the virologic outcomes), because the risk of ascertainment bias is limited. We made a judgment of "low risk of bias" for the remaining studies defined as double-blind. We judged the masking of outcome assessor to treatment allocation at "low risk" of bias for 4 studies in which the assessment was performed by someone not involved in the study (GS 1489, GS1490, SINGLE, SPRING-2) [14–16, 18, 21–24], and at unclear risk of bias for the remaining 3 studies where it was unclear if adequate measures were taken to ensure that the assessors were unaware of treatment allocation. Although all the included studies were sponsored by pharmaceutical companies, our evaluation showed that

Table 1. Main characteristic of included studies.

| Study name | Methods | Interventions | Outcomes data extracted | Notes |
|---|---|---|---|---|
| ARIA [12] | Multicenter, RCT, open-label, non-inferiority phase 3b study | DTG+ ABC and 3TC OD or ATV/r + coformulated TDF and FTC, OD | HIV-1 RNA <50 cps/mL at week 48. Adverse events, adverse events requiring discontinuation, overall rate of discontinuation. Virologic failure. Death | ITT analysis according to the FDA snapshot algorithm. |
| FLAMINGO [13. 19] | Multicenter, RCT, open-label, non-inferiority phase 3b study | DTG or DRV/r, with TDF/FTC or ABC/3TC, OD. | HIV-1 RNA <50 cps/mL at week 48 and 96. Adverse events, adverse events requiring discontinuation, overall rate of discontinuation. Virologic failure. Death | ITT analysis according to the FDA snapshot algorithm Data at 96 wks reported in a separate report[19] |
| GS 1489 [14, 24] | double-blind, multicenter, RCT non-inferiority trial | BIC + FTC and TAF, or coformulated DTG/ABC/3TC with matching placebo, OD for 144 weeks. | HIV-1 RNA <50 cps/mL at week 48 and 96. Adverse events, adverse events requiring discontinuation, overall rate of discontinuation. Virologic failure. Death | ITT analysis according to the FDA snapshot algorithm |
| GS 1490 [15, 23] | double-blind, multicenter, RCT non-inferiority trial | BIC + FTC and TAF, or DTG + FTC and TAF, with matching placebo, OD for 144 weeks. | HIV-1 RNA <50 cps/mL at week 48 and 96. Adverse events, adverse events requiring discontinuation, overall rate of discontinuation. Virologic failure. Death | ITT analysis according to the FDA snapshot algorithm |
| SINGLE [16, 22] | Double-blind, RCT, phase 3 trial | DTG + ABC and 3TC or combination therapy with EFV/TDF/FTC | HIV-1 RNA <50 cps/mL at wks 48 and 96/144. Adverse events, adverse events requiring discontinuation, overall rate of discontinuation. Virologic failure. Death | ITT analysis according to the FDA snapshot algorithm. Data at 96 and 144 wks reported in a separate report[22] |
| SPRING-1 [17, 20] | RCT, dose ranging, phase 2 study | DTG 10, 25 or 50 mg or 600 mg efavirenz, with either TDF/FTC or ABC/3TC | HIV RNA <50 cps at wks 48 and 96. Adverse events, adverse events requiring discontinuation, overall rate of discontinuation. Virologic failure | ITT analysis using the FDA TLOVR approach. Dose but not drug allocation was masked. Data at 96 wks reported in a separate report.[20] |
| SPRING-2 [18, 2] | Double-blind, RCT, phase 3 study | DTG or RTG 400 bid + coformulated TDF/FTC or ABC/3TC | HIV RNA <50 cps at wks 48 and 96. Adverse events, adverse events requiring discontinuation, overall rate of discontinuation. Virologic failure. Death | ITT analysis according to the FDA snapshot algorithm Data at 96 wks reported in a separate report.[21] |

they were of high methodological quality and also excluded the presence of other biases that cannot be explained by standard 'Risk of bias' assessments (eg, deficiencies in the definition of

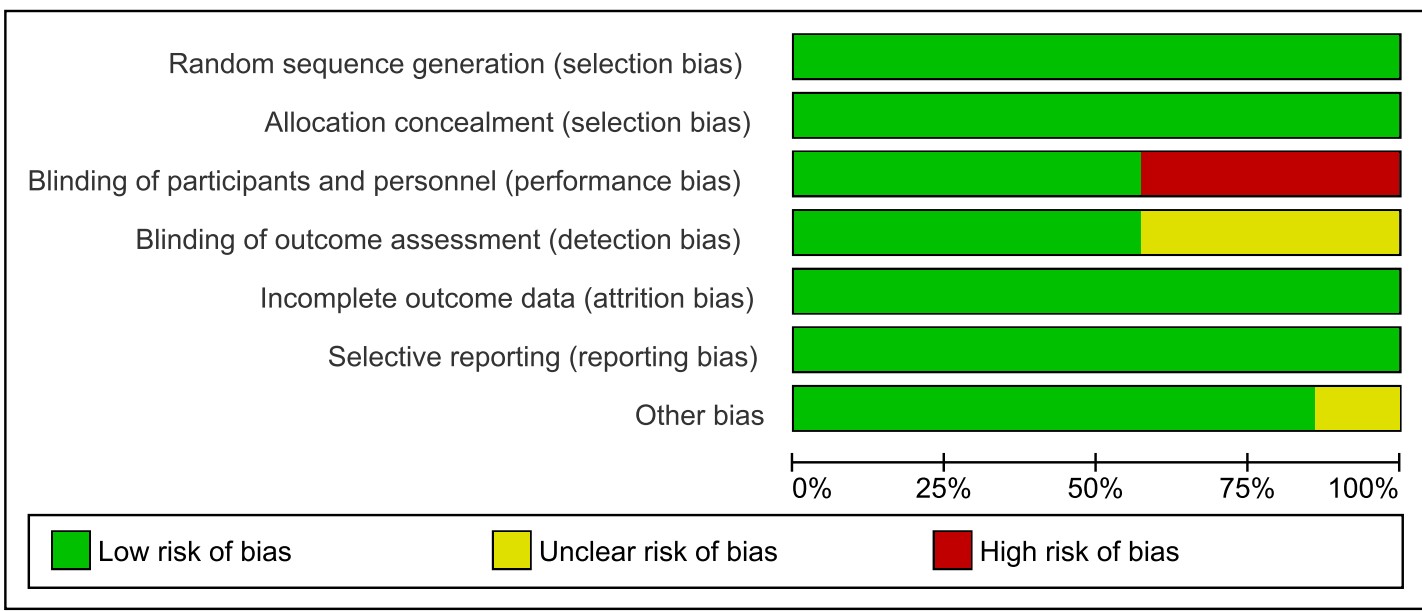

Fig 2. Risk of bias graph: Review authors' judgment about each risk of bias item presented as percentage across all included studies.

| | Random sequence generation (selection bias) | Allocation concealment (selection bias) | Blinding of participants and personnel (performance bias) | Blinding of outcome assessment (detection bias) | Incomplete outcome data (attrition bias) | Selective reporting (reporting bias) | Other bias |
|---|---|---|---|---|---|---|---|
| ARIA | + | + | − | ? | + | + | ? |
| Flamingo | + | + | − | ? | + | + | + |
| GS 1489 | + | + | + | + | + | + | + |
| GS 1490 | + | + | + | + | + | + | + |
| SINGLE | + | + | + | + | + | + | + |
| SPRING 1 | + | + | − | ? | + | + | + |
| SPRING 2 | + | + | + | + | + | + | + |

**Fig 3. Risk of bias summary: Review authors' judgment about each risk of bias item for each included included study.**

patient-relevant endpoints, selection of suitable substances for the control arm, evidence of publication bias); moreover, the protocol of these studies were registered on international registers of clinical trials (clinicalTrials.gov) and results publicly available. Therefore we made a judgment of "low risk of bias" for the domain "other bias.

Funnel plots were not performed because when the number of studies is small (e.g., <10) the plot may not detect publication. [10]

## Effects of interventions

Table 2 summarizes the pooled outcome data and grade assessment for the outcomes in overall analyses and subgroup analyses. We evaluated virologic outcomes, overall rate of discontinuation and/or switching for any cause, adverse events requiring discontinuation of treatment, occurrence of any adverse events, most common adverse events, virologic failure, and death. No AIDS defining conditions were reported across the studies.

**Virologic outcomes.** For most of the virologic outcomes, the GRADE assessment showed high quality of evidence (Tables 2 and 3).Rates of patients with VL <50 copies/ml at 48 weeks were higher in dolutegravir group compared to the alternative agents in the overall comparison (RD, 0.05; 95% CIs, 0.03/0.08, p = 0.0002), in the subgroups of patients with high baseline VL (RD, 0.10; 95% CIs, 0.05/0.15; p< 0.0001), and in the subgroup of patients with low baseline VL (RD, 0.03; 95% CIs, 0.01/0.06; p< 0.01) (Fig 4, Tables 2 and 3).

The results of subgroups analyses according to comparators (other INSTI, PI, efavirenz) are summarized in Table 2 and supplemental Figs 1–4 (S1–S4 Figs). Of note, in the comparison with efavirenz, the effect size was mostly driven mostly by one of the included studies.[16]

At 96 weeks (Table 2, S5 Fig), data from 6 reports (2 versus efavirenz, one versus darunavir, 2 versus bictegravir and one versus raltegravir) show that, compared to controls, an higher proportions of dolutegravir recipients had undetectable viremia (RD 0.06; 95% CI, 0.03/0.10) [19–24].

Analyses according to baseline CD4 values show a significant higher proportions of undetectability in dolutegravir compared to controls in the subgroup of patients with baseline CD4 levels >200 cells/mmc (RD 0.04; 95% CI, 0.01/0.07); in the subgroup of patients with <200 cells/mmc at baseline there was still a higher proportions of undetectability, of borderline significance (RD 0.06; 95% CI, 0.00/0.11; p = 0.05), in dolutegravir compared to controls, (Table 2, S6 Fig). Virologic outcomes from 3 studies reporting results according to the NRTI backbone used are summarized on Table 2 [13, 17, 18].

Protocol- defined virologic failure (generally defined as two consecutive HIV-1 RNA values of at least 50 copies/mL) occurred rarely both in dolutegravir and controls groups (2.5 and 3.3%, respectively) (Table 2). No geno-phenotypic mutation were recorded among dolutegravir recipients; in the control groups, few NNRTI and/or NRTI mutations were recorded in 3 studies [12, 16, 18]. Notably, in the SPRING-2 study one patient in the raltegravir group with baseline plasma HIV-1 RNA of more than 3 million copies/ mL developed both integrase-resistant and NRTI-resistant mutations; phenotype resistance at virologic failure showed a raltegravir fold-change of 34 and a dolutegravir fold-change of 2 [18].

**Other outcomes.** Overall rate of discontinuation and/or switching for any cause (virologic failure, clinical failure, adverse events) was lower in dolutegravir recipients compared to controls (RD, -0.03, 95% Cis, -0.05/-0.01; p = 0.0007; "high certainty of evidence) (Fig 5, Tables 2 and 3). The occurrence of any adverse events and of adverse events requiring discontinuation of treatment was not significantly different in dolutegravir compared to controls (p = 0.84 and

**Table 2. Summary of the pooled outcome data.** Results are provided for all possible comparisons, and for subgroups analyses.

| Outcome or subgroup | Studies (no. pts) | No. with event /Total no. pts: DTG & control | Effect Estimate (RD and 95% CI) | P value | Grades of evidence |
|---|---|---|---|---|---|
| **VL <50 cps/ml, 48 wks** | | | | | |
| **DTG vs all comparators** | 7(4113) | 1880/2110 & 1679/2003 | 0.05 (0.03/0.08) | 0.0002 | High |
| - High baseline VL | 7 (1019) | 440/515 & 382/504 | 0.10 (0.05/0.15) | <0.0001 | High |
| - Low baseline VL | 7 (3094) | 1441/1595 & 1297/1499 | 0.03 (0.01/0.06) | 0.01 | High |
| **DTG vs INTI** | 3(2096) | 958/1051 & 924/1045 | 0.03 (0.00/0.05) | 0.04 | High |
| -High baseline VL | 3(453) | 190/218 & 188/235 | 0.08 (0.01/0.14) | 0.03 | High |
| - Low baseline VL | 3(1643) | 768/833 & 736/810 | 0.01 (-0.01/0.04) | 0.33 | Moderate* |
| **DTG vs PI** | 2 (979) | 420/490 & 376/489 | 0.11 (0.02/0.20) | 0.02 | High |
| -High baseline VL | 2 (257) | 112/130 & 85/127 | 0.20 (0.10/0.30) | <0.0001 | High |
| -Low baseline VL | 2 (722) | 308/360 & 291/362 | 0.05 (-0.02/0.12) | 0.19 | Moderate* |
| **DTG vs NNRTI (efavirenz)** | 2 (1038) | 503/569 & 379/469 | 0.07 (0.03/0.12) | 0.001 | High |
| -High baseline VL | 2 (247) | 138/167 & 109/142 | 0.06 (-0.03/0.15) | 0.21 | Moderate* |
| -Low baseline VL | 2 (729) | 365/402 & 270/327 | 0.08 (0.03/0.13) | 0.002 | High |
| **VL <50 cps/ml, 96 wks** | | | | | |
| DTG vs all comparators | 6(3624) | 1559/1862 & 1357/1762 | 0.06 (0.03/0.10) | 0.0001 | Moderate** |
| **VL <50 cps/ml, 48 wks, per protocol analysis** | | | | | |
| DTG vs all comparators | 6(3689) | 1717/1853 & 1615/1836 | 0.04 (-0.00/0.09) | 0.06 | Low*** |
| **VL <50 cps/ml, 48 wks, according to baseline CD4 count (</> 200 cells/ml)** | | | | | |
| <200 CD4 | 6 (530) | 221/265 & 207/265 | 0.06 (0.00/0.11) | 0.05 | Moderate* |
| >200 CD4 | 6 (3478) | 1515/1690 & 1434/1688 | 0.04 (0.01/0.07) | 0.009 | Moderate** |
| **VL <50 cps/ml, 48 wks, according to NRTI backbone** | | | | | |
| DTG vs controls, ABC/3TC as backbone | 3 (652) | 255/310 & 288/360 | -0.1 (-0.06/0.05 | 0.82 | Moderate* |
| DTG vs controls, TDF/FTC as backbone | 3 (869) | 446/509 & 299/360 | 0.05 (0.00/0.10 | 0.04 | High |
| DTG combined to ABC/3TC | 3 (310) | 255/310 | -0.06/0.00 | 0.06 | Moderate* |
| DTG combined to TDF/FTC | 3 (509) | 446/509 | | | |
| **Overall rate of discontinuation** | | | | | |
| DTG vs all comparators | 7 (4118) | 188/2111 & 257/2007 | -0.03 (-0.05/-0.01) | 0.0007 | High |
| **Adverse events requiring discontinuation** | | | | | |
| DTG vs all comparators | 7 (4117) | 40/2111 & 85/2006 | -0.02 (-0.05/0.00) | 0.10 | Low** |
| **Any Adverse Events** | | | | | |
| All studies | 7 (4117 | 1273/2111 & 1221/2006 | -0.01 (-0.07/0.06 | 0.84 | Very Low$ |
| Double blind studies only | 4 (2301) | 615/1205 & 508/1096 | 0.03 (-0.06/0.12) | 0.46 | Low*** |
| **Most common adverse events** | | | | | |
| -Nausea | | | | | |
| All studies | 7 (4113) | 250/2110 & 190/2003 | 0.03 (-0.01/0.06( | 0.19 | Very Low$ |
| Double blind studies only | 4 (2301) | 179/1206 & 113/1095 | 0.05 (-0.00/0.11) | 0.07 | Very Low$ |
| -Diarrhoea | | | | | |
| All studies | 7 (4122) | 225/2110 & 246/2012 | -0.01 (-0.04/0.02) | 0.46 | Low*** |
| Double blind studies only | 4 (2310) | 139/1206 & 127/1104 | 0.00 (-0.02/0.03) | 0.72 | Moderate* |
| -Insomnia | | | | | |
| All studies | 7 (3933) | 103/1930 & 91/2003 | 0.01 (-0.01/0.03) | 0.46 | Very Low § |
| Double blind studies only | 4 (2121) | 58/1026 & 51/1095 | 0.01 (-0.03/0.06) | 0.61 | Low*** |
| -Psychiatric disorders | | | | | |
| All studies | 4 (2629) | 114/1315 & 115/1314 | -0.00 (-0.04/0.04 | 0.95 | Very Low$ |
| Double blind studies only | 1 (822) | 34/411 & 33/411 | 0.00 (-0.03/0.04) | 0.90 | Low*** |

(*Continued*)

**Table 2.** (Continued)

| Outcome or subgroup | Studies (no. pts) | No. with event /Total no. pts: DTG & control | Effect Estimate (RD and 95% CI) | P value | Grades of evidence |
|---|---|---|---|---|---|
| **Virologic failure (Protocol-defined)** | 7 (4119) | 53/2115 & 68/2004 | -0.01 (-0.02/0.01) | 0.39 | Low** |
| **Death** | 7 (4117) | 5/2111 & 5/2006 | -0.00 (-0.00/0.00) | 0.81 | Low§§ |

Notes:

Abbreviations: DTG, dolutegravir; ABC, abacavir;: 3TC, lamivudine; FTC, emtricitabine; TDF, tenofovir disoproxil fumarate; NNRI, nonnucleoside reverse

transcriptase inhibitor; PI, protease inhibitor, INTI, integrase strand transfer inhibitor; cps, copies; VL, Viral Load; wks, weeks

*Downgraded once for imprecision (95% CI includes line of no effect)

** Downgraded once for heterogeneity

*** Downgraded twice for imprecision and heterogeneity

§ Downgraded for imprecision, heterogeneity and risk of bias (performance and detection bias) in open label studies

§§ downgraded for imprecision and indirectness

p = 0.10, respectively). Most common adverse events related to study treatment were nausea, diarrhea, insomnia and psychiatric disorders, not significantly different between dolutegravir recipients and controls, both in the whole analyses and in subgroup analyses of double blind studies (Table 2, S7 Fig). The certainty of evidence for adverse events outcomes was low or very low in the majority of comparisons (downgraded for imprecision, heterogeneity and/or performance and detection bias).

## Discussion

For the treatment of HIV infected naive patients current guidelines endorse an ARV regimen generally consisting of two NRTI administered in combination with a third active ARV drug from one of three drug classes: INSTI, NNRTI, or boosted PI [2, 25]. For most patients, initial therapy should be with two NRTIs combined with an INSTI; in some individuals, a combination of an NNRTI or boosted PI should be considered. Basing on recent guidelines for the use of antiretroviral agents in adults and adolescents with HIV, "the choice between an INSTI, PI, or NNRTI as the third drug in an initial ARV regimen should be guided by the regimen's efficacy, barrier to resistance, adverse effects profile, convenience, comorbidities, concomitant medications, and the potential for drug-drug interactions" [2]. Dolutegravir in combination with NRTIs is one of the recommended regimens in antiretroviral therapy-naïve patients, but others INSTI, such as raltegravir and, more recently, bictegravir, can be considered [2, 25]. In clinical trials, these INSTI-containing regimens were highly effective and have relatively infrequent adverse effects and few drug interactions. In several head-to-head comparisons between boosted PI-containing regimens and INSTI-containing regimens, the INSTI was better tolerated and caused fewer treatment discontinuations [13, 26]. On the basis of data from in vitro studies and clinical trials in ART-naive patients, it is anticipated that, like dolutegravir, bictegravir has a high barrier to resistance. However, clinical data and experience with bictegravir are relatively limited right now [2]. Although the success of ART is beyond question, several issues related to safety and efficacy still persist. Pre-treatment viral load level is an important factor in the selection of an initial antiretroviral regimen because several antiretroviral drug regimens have been associated with poorer responses and slower rate of achieving viral suppression in patients with high baseline VL [2, 3].

Numerous RCTs were conducted to evaluate dolutegravir containing ART compared to other ART regimens in naïve patients, and these have been the objectives of systematic reviews [7, 27–30]. Since then, however, new evidence is accumulating, and a new review is required

**Table 3. Summary of findings table.** ART with DTG compared with ART with other core agents for HIV-1 infected naive patients.

Patient or population: treatment-naive patients with HIV infection
Settings: outpatients
Intervention: DTG in combination with 2 NRTI
Comparison: PI (boosted DRV, ATV), or NNRTI (EFV), or INSTI (RAL, BIC) in combination with 2 NRTI

| Outcomes | Illustrative comparative risks * (95% CI) | | Relative effect: (95% CI) | No of Participants (studies) | Quality of the evidence (GRADE)** | Comments |
|---|---|---|---|---|---|---|
| | Assumed risk | Corresponding risk | | | | |
| | CONTROLS (ALL) | DTG | | | | |
| Virologic Outcomes: % pts with VL <50 cps/ml at 48 wks | | | | | | |
| all pts, regardless to baseline VL | 83.8% (1679/2003) | 89.2% (86.3–90.5%) | RD, 0.05 (0.03/0.08) | 4113 (7) | ⊕⊕⊕⊕ high *** | Starting treatment with DTG containing ART has an increased likelihood of achieving VL <50 cps/ml at 48 wks compared to the alternative treatments |
| baseline VL >100.000 cps/ml | 75.7% (382/504) | 83.2% (79.4–87.0%) | RD, 0.10 (0.05/0.15) | 1019 (7) | ⊕⊕⊕⊕ high | The average benefit is particularly evident in those with high baseline VL (+10%, CIs +5/+15%) |
| baseline VL <100.000 cps/ml | 86.5% (1297/1499) | 89.0% (87.3–91.7%) | RD, 0.03 (0.01/0.06) | 3094 (7) | ⊕⊕⊕⊕ high | The benefit includes also pts with low screening VL |
| | INSTI (BIC, RAL) | DTG | | | | |
| all pts, regardless to baseline VL | 88.4% (924/1045) | 91.0% (88.4–92.8%) | RD 0.03 (0.00/0.05) | 2096 (3) | ⊕⊕⊕⊕ high | Starting treatment with DTG containing ART has an increased likelihood of achieving VL <50 cps/ml at 48 wks compared to other INSTI |
| baseline VL >100.000 cps/ml | 80.0% (188/235) | 86.4% (80.8–91.2%) | RD 0.08 (0.01/0.14) | 453 (3) | ⊕⊕⊕⊕ high | The average benefit is particularly evident in those with high baseline VL (+8%, CIs +1/+14) |
| baseline VL <100.000 cps/ml | 90.8% (736/810) | 91.7% (89.9–94.4%) | RD 0.01 (-0.01/0.04) | 1643 (3) | ⊕⊕⊕⊖§ moderate | On average, it is unclear whether or not use of DTG compared to other INSTI increases rates of pts with undetectable VL. |
| | NNRTI (EFV) | DTG | | | | |
| all pts, regardless to baseline VL | 80.8% (379/469) | 86.4% (83.2–90.4%) | RD 0.07 (0.03/0.12) | 1038 (2) | ⊕⊕⊕⊕ high | starting treatment with DTG containing ART has an increased likelihood of achieving VL <50 cps/ml at 48 wks compared to EFV |
| baseline VL >100.000 cps/ml | 76.7% (109/142) | 81.3% (74.4/88.2%) | RD 0.06 (-0.03/0.15) | 247 (2) | ⊕⊕⊕⊖§ moderate | On average, it is unclear whether or not use of DTG compared to EFV increases rates of pts with undetectable VL in subgroup of pts with high baseline VL |
| baseline VL <100.000 cps/ml | 82.5% 8270/327) | 89.1 (84.9–93.2%) | RD 0.08 (0.03/0.13) | 729 (2) | ⊕⊕⊕⊕ high | starting treatment with DTG containing ART has an increased likelihood of achieving VL <50 cps/ml compared to EFV in subgroup of pts with low baseline VL |
| | PI (DRV, ATV) | DTG | | | | |
| all pts, regardless to baseline VL | 76.8% (376/489) | 85.2% (78.3–92.1%) | RD 0.11 (0.02/0.20) | 979 (2) | ⊕⊕⊕⊕ high | starting treatment with DTG containing ART has an increased likelihood of achieving VL <50 cps/ml at 48 wks compared to PI |
| baseline VL >100.000 cps/ml | 66.9% (85/127) | 77.2% (73.5–86.9) | RD 0.20 (0.10/0.30) | 257 (2) | ⊕⊕⊕⊕ high | The average benefit is particularly evident in those with high baseline VL (+20%, 95 CIs +10/+30) |
| baseline VL <100.000 cps/m | 80.3% (291/362) | 84.3% (78.7–89.9%) | RD 0.05 (-0.02/0.12) | 722 (2) | ⊕⊕⊕⊖§ moderate | On average, it is unclear whether or not use of DTG compared to PIs increases rates of pts with undetectable VL in subgroup of pts with low baseline VL |
| Overall rate of discontinuation of treatment in DTG recipients and controls (INSTI, PI, EFV) at 48 wks. | | | | | | |
| All pts. | 12.8% (257/2007) | 9.8% (7.8/11.8%) | RD -0.03 (-0.05/-0.01) | 4118 (7) | ⊕⊕⊕⊕ high | Rates of interruption of treatment for any reason (virologic failure, clinical failure, adverse events) were significantly lower in DTG recipients compared to controls |

Footnotes:

*The **assumed risk** is the mean control group risk of having VL <50 cps/ml at 48 wks across studies; the **corresponding risk** (and its 95% CI) is based on the assumed risk in the comparison group and the **relative effect** of the intervention (and its 95% CI).

** GRADE Working Group grades of evidence

**High quality:** Further research is very unlikely to change our confidence in the estimate of effect.

**Moderate quality:** Further research is likely to have an important impact on our confidence in the estimate of effect and may change the estimate.

**Low quality:** Further research is very likely to have an important impact on our confidence in the estimate of effect and is likely to change the estimate.

**Very low quality:** We are very uncertain about the estimate.

***Despite the fact that 3 of the included studies were judged at high risk of performance bias (open label), we judged this as high-certainty evidence because masking has limited importance for the virologic outcomes, because the risk of ascertainment bias is limited.

§ Downgraded once for imprecision (95%CI includes line of no effect)

Abbreviations: pts, patients; VL, viral load; cps, copies; wks, weeks; PI, protease inhibitors; INSTI, integrase strand transfer inhibitors; NNRTI, non-nucleoside reverse transcriptase inhibitor; NRTI, nucleoside reverse transcriptase inhibitor; EFV, efavirenz; DTG, dolutegravir; BIC, bictegravir; RAL, raltegravir; DRV, darunavir; ATV, atazanavir. CI, Confidence interval; RD, Risk Difference.

to shed more light on safety and effectiveness issues related to ART regimens containing or not dolutegravir.

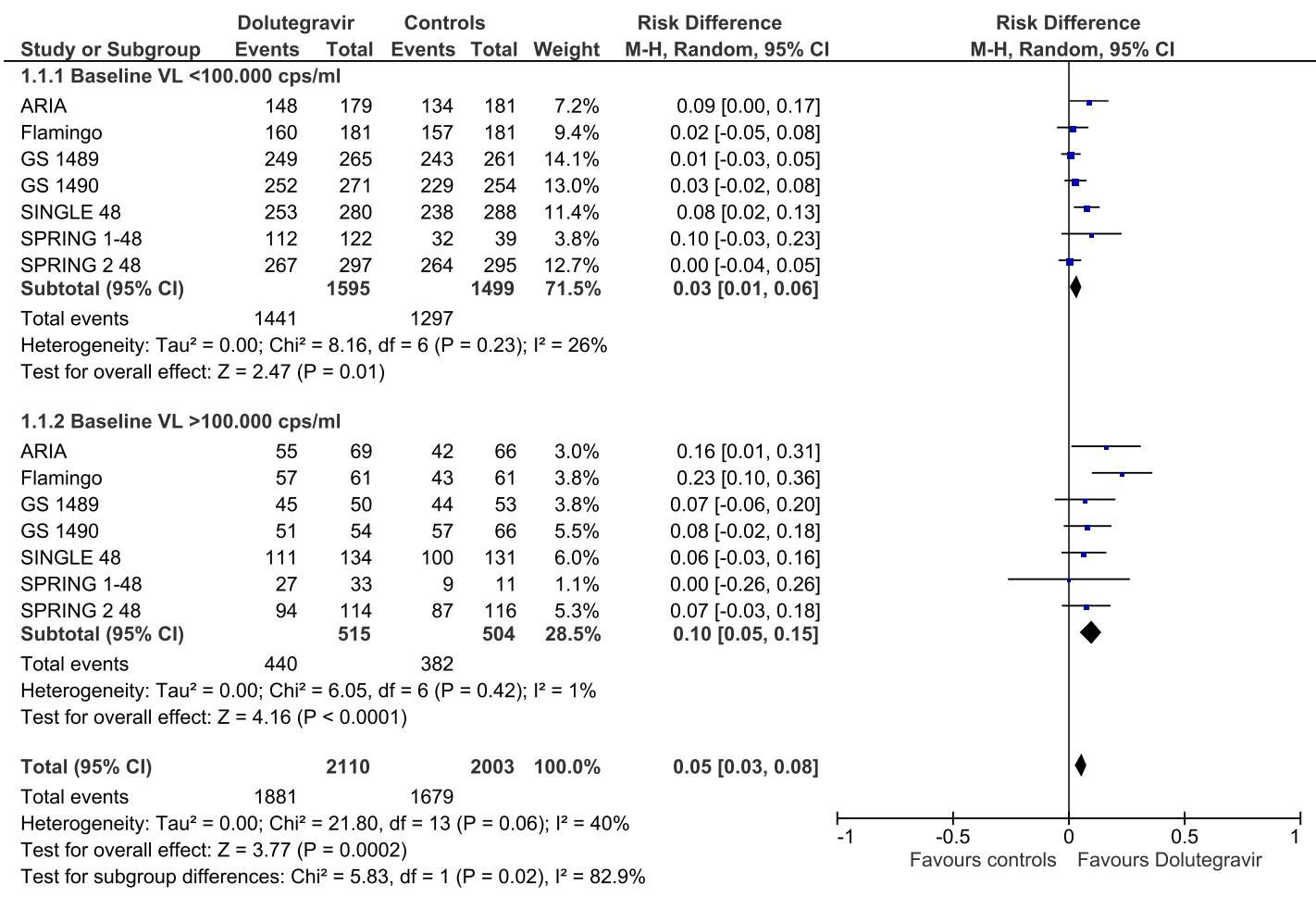

**Fig 4. Forest plot of comparison.** Doultegravir vs comparators, outcome: VL<50 copies at 48 weeks.

In this systematic review we have included seven primary studies, corresponding to 13 reports, evaluating the efficacy and safety of ART regimens containing dolutegravir or others core agents (PIs, NNRTI or other INSTI), both in combination with a dual nucleoside

**Fig 5. Forest plot of comparison.** Dolutegravir vs comparators, outcome: Overall rate of discontinuation.

backbone (TDF/FTC, TAF/FTC or ABC/3TC) in naïve patients. Bias assessment using Cochrane methodology showed that the studies analyzed had few methodological limitations. Only four of the seven included studies were blind. However, the primary outcome of the analysis and its measurement (viral load) is not likely to be influenced by lack of blinding; on the other hand, blinding could have been important for assessment of subjective outcomes such as adverse events.

The primary endpoint of the included studies was the proportion of patients with HIV-1 RNA concentration lower than 50 copies/ ml. Our systematic review and meta-analysis demonstrate convincing evidence that starting ART with dolutegravir in naïve patients has important clinical implications. Actually, we observe an increased likelihood (+5%; 95% CIs, +3/ +8%) of achieving VL<50 copies/ml at 48 weeks compared to the alternative agents (boosted atazanavir or darunavir, efavirenz, raltegravir or bictegravir). The average benefit is particularly evident (+10%; 95%CIs, +5%/+15%) in the subgroup of patients with high baseline VL: in this subgroup of patients the superior activity of dolutegravir is consistent for the comparison with other INSTI (+8%; 95% CIs, +1/+14%) and with boosted PIs (+20%; 95% CIs, +10/ +30%). Basing on GRADE assessment, all these comparisons were graded as "high certainty of evidence". An assessment of high-certainty evidence means that further research is very unlikely to change our confidence in the estimates of the effect. At 96 weeks dolutegravir recipients still had higher proportion of undetectable viremia (+ 6%; 95% CIs, +3/+10%) compared to controls (efavirenz, darunavir, bictegravir or raltegravir) (moderate certainty of evidence due to heterogeneity).Overall rate of discontinuation and/or switching for any cause (virologic failure, clinical failure, adverse events) at 48 weeks were lower in dolutegravir group compared to controls (-3%; 95% CIs, -5/-1%) (high certainty of evidence).

Data on adverse events (any adverse events, adverse events requiring discontinuation, most common adverse events) produced low or very low certainty evidence, due to imprecision, heterogeneity and/or risk of bias in open label studies. An assessment of low-certainty evidence means that our confidence in the effect estimate is limited, and the true effect may be substantially different from the estimate of the effect. There were no statistically significant differences in the occurrence of any adverse events, adverse events requiring discontinuation and most common adverse events in dolutegravir group compared to controls, in the overall analysis and in the analysis of double-blind studies only. Most common adverse events reported in both groups were nausea, diarrhea, insomnia and psychiatric disorders.

Protocol- defined virologic failure occurred rarely both in dolutegravir and controls groups. Notably, in trials included in the current review no geno-phenotypic mutations and mutations that confer dolutegravir resistance have been recorded, which suggests that dolutegravir, like bictegravir, has a higher barrier to resistance than raltegravir and elvitegravir. Resistance to dolutegravir and bictegravir has not been reported in clinical trials when these drugs are used as part of initial triple-drug ART, and only rarely in treatment-experienced patients receiving a dolutegravir-containing regimen [13, 15, 16, 18, 31–33]. Combinations of INSTI resistance mutations selected by prolonged exposure to raltegravir or elvitegravir in the setting of treatment failure can, however, result in cross-resistance to dolutegravir and bictegravir [34–36].

In all the included studies, the experimental intervention was compared to an established standard-of-care treatment, as endorsed by current guidelines. Dolutegravir and comparators were administered with fixed-dose combination of TDF/FTC, or TAF/FTC, or ABC/3TC Screening for hypersensitivity reaction was performed in all the studies included in the current review, with the exception of one study where TAF/FTC was the only NRTI combination administered [15].

The strenghts of the current systematic review is that it address timely and relevant clinical question, such as that related to the virologic efficacy of ART regimens in naïve patients

according to baseline VL levels, and includes studies which were not included in previously published reviews and meta analyses. The limitation is that health outcome measures of quality of life were rarely reported in the included studies, and we could not assess treatment satisfaction.

In summary, we found high quality evidence related to the higher virologic efficacy of dolutegravir-containing ART compared to other non-dolutegravir containing ART in naïve patients. For other outcomes such as occurrence of adverse events the available evidence is not conclusive. Although the randomized clinical trial setting would provide the least biased approach to assessing outcomes, blinded outcome assessment is crucial for outcomes such as adverse events, which are inherently subjective. Not all the included studies were blinded, so the risk of detection or observer bias cannot be excluded.

## Conclusion

Starting treatment in naive patients with dolutegravir containing ART has an increased likelihood of achieving viral suppression in the comparison with non-dolutegravir containing ART. The average benefit is particularly evident in those with high baseline VL (eg, >100,000 copies/ml), and it is consistent in trials comparing dolutegravir to other INSTI or to boosted PIs.

## Supporting information

**S1 Checklist. PRISMA 2009 checklist.**
(PDF)

**S1 Fig. Forest plot of comparison: DTG vs INTI, outcome: VL<50 copies/ml at 48 wks.**
(EPS)

**S2 Fig. Forest plot of comparison: DTG vs PI, outcome: VL <50 copies/ml at 48 wks.**
(EPS)

**S3 Fig. Forest plot of comparison: DTG vs BIC, outcome: VL < 50copies/ml at 48 wks.**
(EPS)

**S4 Fig. Forest plot of comparison: DTG vs EFV, outcome: VL<50 copies/ml at 48 wks.**
(EPS)

**S5 Fig. Forest plot of comparison: Dolutegravir vs comparators, outcome: VL<50 copies/ml at 96 wks.**
(EPS)

**S6 Fig. Forest plot of comparison: DTG vs other according to baseline CD4 (cut off 200 CD4/ml), outcome: % <50 copies/ml at 48 wks.**
(EPS)

**S7 Fig. Forest plot of comparison: Dolutegravir vs comparators, outcome: Most common adverse effects.**
(EPS)

## Acknowledgments

Part of this study was presented at the 27[th] Conference on the Italian Society of Infectious and Tropical Diseases (SIMIT), December 2–5 2018, Turin, Italy.

## Author Contributions

**Conceptualization:** Mario Cruciani, Saverio G. Parisi.

**Data curation:** Mario Cruciani, Saverio G. Parisi.

**Formal analysis:** Mario Cruciani.

**Funding acquisition:** Saverio G. Parisi.

**Investigation:** Mario Cruciani, Saverio G. Parisi.

**Methodology:** Mario Cruciani.

**Software:** Mario Cruciani.

**Supervision:** Mario Cruciani, Saverio G. Parisi.

**Validation:** Mario Cruciani, Saverio G. Parisi.

**Visualization:** Mario Cruciani.

**Writing – original draft:** Mario Cruciani, Saverio G. Parisi.

**Writing – review & editing:** Mario Cruciani.

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
