## [Decision Letter · Decision Letter 0]

24 Jul 2019

PONE-D-19-17880

Dolutegravir based antiretroviral therapy compared to other combined antiretroviral regimens for the treatment of HIV-infected naive patients: a systematic review and meta-analysis.

PLOS ONE

Dear Dr. Cruciani,

Thank you for submitting your manuscript to PLOS ONE. After careful consideration, we feel that it has merit but does not fully meet PLOS ONE’s publication criteria as it currently stands. Therefore, we invite you to submit a revised version of the manuscript that addresses the points raised during the review process.

The reviewer(s) and I have evaluated the paper. After full consideration of the comments, I believe that the paper could be of interest to PLOS ONE readers but a number of questions must be addressed and changes  made to render this manuscript suitable for publication, See comments by Reviewer #1.

We would appreciate receiving your revised manuscript by the next 12 weeks. To enhance the reproducibility of your results, we recommend that if applicable you deposit your laboratory protocols in protocols.io, where a protocol can be assigned its own identifier (DOI) such that it can be cited independently in the future. For instructions see: http://journals.plos.org/plosone/s/submission-guidelines#loc-laboratory-protocols

We look forward to receiving your revised manuscript.

Kind regards,

Giuseppe Vittorio De Socio, MD, PhD

Academic Editor

PLOS ONE

Journal Requirements:

4. Please confirm that you have included all items recommended in the PRISMA checklist including an assessment of publication bias using graphical methods (e.g. Funnel plot) and statistical methods (e.g. Egger’s test) as appropriate; details of reasons for study exclusions in the PRISMA flowchart and number of studies excluded for each reason; and the full electronic search strategy used to identify studies with all search terms and limits for at least one database.

5. We noticed you have some minor occurrence of overlapping text with the following previous publication(s), which needs to be addressed:

https://aidsinfo.nih.gov/contentfiles/lvguidelines/AdultandAdolescentGL.pdf

https://academic.oup.com/jid/article/218/5/673/4956225

The text that needs to be addressed is in the Discussion section.

In your revision ensure you cite all your sources (including your own works), and quote or rephrase any duplicated text outside the methods section. Further consideration is dependent on these concerns being addressed.

6.  Thank you for stating the following in the Competing Interests section:

MC has received honoraria as a speaker and/or advisor from  Abbott, Bayer, Cephalon, Gilead, Novartis and ViiV Healthcare, SGP  has received  research grants from Gilead Sciences, ViiV Healthcare, Abbott, MS&D, and  honoraria as a speaker from MS&D, Gilead Sciences, ViiV Healthcare, Abbvie, Janssen. These activities were not related to their work with this review

Reviewers' comments:

Reviewer's Responses to Questions

**Comments to the Author**

1. Is the manuscript technically sound, and do the data support the conclusions?

Reviewer #1: Partly

Reviewer #2: Yes

2. Has the statistical analysis been performed appropriately and rigorously? 

Reviewer #1: Yes

Reviewer #2: Yes

3. Have the authors made all data underlying the findings in their manuscript fully available?

Reviewer #1: Yes

Reviewer #2: Yes

4. Is the manuscript presented in an intelligible fashion and written in standard English?

Reviewer #1: Yes

Reviewer #2: Yes

5. Review Comments to the Author

Reviewer #1: There are several mislabeled items in the plots. Please mention if the authors performed tests for multiplicity.

Reviewer #2: The Authors performed a systematic review and meta-analysis of RCTs comparing dolutegravir-containing ART to non-dolutegravir containing ART in HIV-infected naive patients.The manuscript is well written and represents a very comprehensive analysis on this topic. The methodology is adequate and updated.

I have just one suggestion. Since the use of patients reported outcomes represent a more useful tool compared to clinician report of adverse events and they can explore outcomes such as patients' satisfaction with treatment, and considering the high virological response rates reported in some comparison between INSTI-based regimens in particular, it could be useful to include, when available, PROs as specific outcomes in the metanalysis (Whol, The Patient - Patient-Centered Outcomes Research, 2018).

6. PLOS authors have the option to publish the peer review history of their article (what does this mean?). If published, this will include your full peer review and any attached files.

Reviewer #1: No

Reviewer #2: No

---

## [Author Response · Author response to Decision Letter 0]

21 Aug 2019

Journal Requirements:

AU: OK

AU: OK

AU: as suggested, we have removed the sentence, and specified that funnel plots were not performed because the low number of studies (lines 199-200)

4. Please confirm that you have included all items recommended in the PRISMA checklist including an assessment of publication bias using graphical methods (e.g. Funnel plot) and statistical methods (e.g. Egger’s test) as appropriate; details of reasons for study exclusions in the PRISMA flowchart and number of studies excluded for each reason; and the full electronic search strategy used to identify studies with all search terms and limits for at least one database.

AU: done

5. We noticed you have some minor occurrence of overlapping text with the following previous publication(s), which needs to be addressed:

https://aidsinfo.nih.gov/contentfiles/lvguidelines/AdultandAdolescentGL.pdf

https://academic.oup.com/jid/article/218/5/673/4956225

The text that needs to be addressed is in the Discussion section.

AU: The first sentence is from DHHS “the choice between an INSTI, PI, or NNRTI as the third drug in an initial ARV regimen should be guided by the regimen’s efficacy, barrier to resistance, adverse effects profile, convenience, comorbidities, concomitant medications, and the potential for drug-drug interactions” We have inserted the whole paragraph in brackets, and specified that the sentence is based on the DHHS guidelines

The second is “result in cross-resistance to dolutegravir and bictegravir. The reference you mention is now quoted (no. 36, Kuritzkes DR. Resistance to Dolutegravir—A Chink in the Armor? The Journal of Infectious Diseases, 2018, 218: 673–675)

In your revision ensure you cite all your sources (including your own works), and quote or rephrase any duplicated text outside the methods section. Further consideration is dependent on these concerns being addressed.

6. Thank you for stating the following in the Competing Interests section:

MC has received honoraria as a speaker and/or advisor from Abbott, Bayer, Cephalon, Gilead, Novartis and ViiV Healthcare, SGP has received research grants from Gilead Sciences, ViiV Healthcare, Abbott, MS&D, and honoraria as a speaker from MS&D, Gilead Sciences, ViiV Healthcare, Abbvie, Janssen. These activities were not related to their work with this review

AU: we have updated this as requested

5. Review Comments to the Author

Reviewer #1: There are several mislabeled items in the plots. Please mention if the authors performed tests for multiplicity.

AU: as we have specified below (comments), we have made consistent legends of figures and graph. We did not perform tests for multiplicity

Reviewer #2: The Authors performed a systematic review and meta-analysis of RCTs comparing dolutegravir-containing ART to non-dolutegravir containing ART in HIV-infected naive patients.The manuscript is well written and represents a very comprehensive analysis on this topic. The methodology is adequate and updated.

I have just one suggestion. Since the use of patients reported outcomes represent a more useful tool compared to clinician report of adverse events and they can explore outcomes such as patients' satisfaction with treatment, and considering the high virological response rates reported in some comparison between INSTI-based regimens in particular, it could be useful to include, when available, PROs as specific outcomes in the metanalysis (Whol, The Patient - Patient-Centered Outcomes Research, 2018).

AU: Thank you for this interesting observation, Unfortunately, as we have now specified in lines 369-371, measures of quality of life were rarely reported in the included studies, and we could not assess treatment satisfaction.

PONE-D-19-17880 

Dolutegravir based antiretroviral therapy compared to other combined antiretroviral regimens for the treatment of HIV-infected naive patients: a systematic review and meta-analysis

Comments: 

Define “naïve” in the Introduction

AU: Done (lineS 64-65)

There are only two authors. Was there an adjudicator in the data extraction process in case of disagreements?

AU: as specified on lines 110-111, agreement was achieved by discussion

90 Assessment of risk of bias and Heterogeneity

Below line 90

Put here gradings of heterogeneity as in I2 = 0% indicating no heterogeneity

AU: Actually, the interpretation of I2 values is more complicated, and can be misleading; often, threshold of interpretation are: ≤25, low heterogeneity; >50, high heterogeneity (now specified in the text, line 94), but these values are completely arbitrary. So, as for other systematic reviews of our groups, we decided to use fixed effect model when I2 =0 and when values of I² are greater than zero, both fixed and random effects analyses were performed (see sentences on line 140-2 “We used the fixed-effect model when the I2 was=0,….”)

Restrict mentioning RevMan more than twice

AU: REV MAN is mentioned on line 132 and 145

Did the authors perform tests for heterogeneity? 

AU: yes (see lines 118-121)

Strategy for data synthesis

104 We used aggregate data. The analysis has been conducted

Substitute the underlined with “was”

AU: done

107 random effects analyses are undertaken and any differences

Substitute the underlined with “performed”

AU: done

109 pre-specified subgroup analysis. We used risk difference as measure of effect. Review Manager 5.3 was used to analyze the data.

RevMan again?

AU: yes, for the second time

Table 2 . legend

Include RD

AU: done

203 adverse events and of adverse events requiring discontinuation of treatment was not significantly different in dolutegravir

What is the p-value for the “not significantly different “?

AU: the p values for any adverse events and adverse events requiring discontinuation have bee added to the text (line 273)

Kindly organize your Discussion so as to indicate summary of effects, clinical implications

Also mention of strengths and limitations of this study.

AU: summary effects and clinical implications are discussed (lines 321-348) and strengths and limits are now mentioned (lines 364-9)

156 Table 2 . Summary of the pooled outcome data. Results are provided for a 156 ll possible comparisons, and for

157 subgroups analyses

In column 5 of the P-values, the authors present several significant values

Did the authors correct for multiple comparisons?

AU: no, we did not.

The favours labels are opposite at the x-axis in Figures 3 and 5

Is this correct? 

AU: yes, it is correct. In fig 4 thr RD is >0, which means that more people in DTG had VL<50 copies, a favourable outcome resulting in effect estimates to the right of the vertical line. In fig. 5 the RD is less than 0, which means that less people in DTG has treatment discontinuation, a favourable outcome resulting in effect estimates to the left of the vertical line

Suppl figures 1 and 2

Need to label DTG in legend

AU: we have made consistent legends of figures and graphs, using the entire words for dolutegravir and comparators

Supl Fig 4. Forest plot of comparison: DTG vs EFV, outcome: VL<50 48 wks

SINGLE 48 contributed 57.7% to the pooled RD, mention this in the text because it is an imbalance in weight contribution

AU: this is now mentioned in the text (lines 236-7)

Supl Fig 5. Forest plot of comparison: Dolutegravir vs comparators, outcome: VL<50 96 wks

The authors used random-effects here, why not fixed?

AU:because I2=31, and we chosed, starting from the protocol, to present the result of the fixed effect model when I2 =0

Supl Fig 6. Forest plot of comparison: DTG vs other according to baseline CD4 (cut off 200 CD4/ml) ,

outcome: % <50.

The authors used random-effects here even if the 9.3.1 plot is I2 = 0%. Please justify

AU: we used random effects model because in one of the subgroup there was evidence of heterogeneity; however, we performed analysis also using the fixed effect, and the results were much the same

Supl Fig 7. Forest plot of comparison: Dolutegravir vs comparators, outcome: Most common adverse

effects. Is the label favouring Dolutegravir same is DTG?

AU: as for fig 5, a RD <0 is a favourable outcome for AE. The 95 % CIs cross the vertical line for all the outcome analyzed

---

## [Editor Report · Decision Letter 1]

26 Aug 2019

Dolutegravir based antiretroviral therapy compared to other combined antiretroviral regimens for the treatment of HIV-infected naive patients: a systematic review and meta-analysis.

PONE-D-19-17880R1

Dear Dr. Cruciani,

We are pleased to inform you that your manuscript has been judged scientifically suitable for publication and will be formally accepted for publication once it complies with all outstanding technical requirements.

With kind regards,

Giuseppe Vittorio De Socio, MD, PhD

Academic Editor

PLOS ONE
---

## [Editor Report · Acceptance letter]

29 Aug 2019

PONE-D-19-17880R1 

Dolutegravir based antiretroviral therapy compared to other combined antiretroviral regimens for the treatment of HIV-infected naive patients: a systematic review and meta-analysis. 

Dear Dr. Cruciani:

I am pleased to inform you that your manuscript has been deemed suitable for publication in PLOS ONE. Congratulations! Your manuscript is now with our production department. 

With kind regards,

on behalf of

Dr. Giuseppe Vittorio De Socio 

Academic Editor

PLOS ONE